# Cascade of neural processing orchestrates cognitive control in human frontal cortex

Hanlin Tang[1,2], Hsiang-Yu Yu[3,4], Chien-Chen Chou[3,4], Nathan E Crone[5], Joseph R Madsen[6], William S Anderson[7], Gabriel Kreiman[1,2,8]*

[1]Program in Biophysics, Harvard University, Boston, United States; [2]Department of Ophthalmology, Boston Children's Hospital, Harvard Medical School, Boston, United States; [3]Department of Neurology, Taipei Veterans General Hospital, Taipei, Taiwan; [4]National Yang-Ming University, Taipei, Taiwan; [5]Department of Neurology, Johns Hopkins School of Medicine, Baltimore, United States; [6]Department of Neurosurgery, Boston Children's Hospital, Harvard Medical School, Boston, United States; [7]Department of Neurosurgery, Johns Hopkins Medical School, Baltimore, United States; [8]Center for Brain Science, Harvard University, Boston, United States

**Abstract** Rapid and flexible interpretation of conflicting sensory inputs in the context of current goals is a critical component of cognitive control that is orchestrated by frontal cortex. The relative roles of distinct subregions within frontal cortex are poorly understood. To examine the dynamics underlying cognitive control across frontal regions, we took advantage of the spatiotemporal resolution of intracranial recordings in epilepsy patients while subjects resolved color-word conflict. We observed differential activity preceding the behavioral responses to conflict trials throughout frontal cortex; this activity was correlated with behavioral reaction times. These signals emerged first in anterior cingulate cortex (ACC) before dorsolateral prefrontal cortex (dlPFC), followed by medial frontal cortex (mFC) and then by orbitofrontal cortex (OFC). These results disassociate the frontal subregions based on their dynamics, and suggest a temporal hierarchy for cognitive control in human cortex.

*For correspondence: gkreiman@gmail.com

**Competing interests:** The authors declare that no competing interests exist.

## Introduction

Flexible control of cognitive processes is fundamental to daily activities, including the execution of goal-directed tasks according to stimulus inputs and context dependencies. An important case of cognitive control arises when input stimuli elicit conflicting responses and subjects must select the task-relevant response despite competition from an often stronger but task-irrelevant response (*Miller, 2000*; *Miller and Cohen, 2001*). A canonical example of this type of conflict is the Stroop task: subjects are asked to name the font color of a word where the semantic meaning conflicts with the color signal (e.g. the word 'red' shown in green versus red). Such incongruent inputs lead to longer reaction times, attributed to weaker signals (color processing) that must be emphasized over the automatic processing of word information (*Stroop, 1935*). The Stroop task is frequently used in cognitive neuroscience and clinical psychology and forms the foundation for theories of cognitive control.

Neurophysiological, neuroimaging, and lesion studies have ascribed a critical role in cognitive control to networks within frontal cortex (*Miller, 2000*; *Miller and Cohen, 2001*), yet the neural circuit dynamics and mechanisms responsible for orchestrating control processes remain poorly understood. Lesion studies (*Cohen and Servan-Schreiber, 1992*; *Perret, 1974*), human neuroimaging measurements (*Egner and Hirsch, 2005a*; *MacDonald, 2000*), and macaque single unit recordings (*Johnston et al., 2007*) implicate the dorsolateral prefrontal cortex (dlPFC) in providing top-down

**eLife digest** The brain adapts to control our behavior in different ways depending on the specific situation, which is particularly useful when deciding how to interpret conflicting sets of information. The 'Stroop task' is a classic demonstration of this process. In this task, individuals are shown words where the color and the meaning of the text conflict – for example, the word 'green' is written in blue. When asked what the color of the text is, individuals must suppress the instinct to read the word. This causes them to make more mistakes and take longer to decide on an answer than when they perform the same task using words that have no conflict (for example, when "red" is written in red).

Previous work has suggested that several regions within part of the brain called the frontal cortex play a role in this cognitive control process. However, the relative contributions of each of these regions, and the order in which they are activated, remain unclear. This is in part due to the fact that accurately measuring the electrical activity of the frontal cortex requires implanting electrodes into the brain.

Tang et al. took advantage of a rare opportunity to record this activity from a group of patients who had electrodes implanted in their frontal cortex to treat epilepsy. The electrical signals recorded by these electrodes as the subjects performed the Stroop task revealed that four regions in the frontal cortex altered their activity during trials where the color and the meaning of a word conflicted. These responses corresponded with the subject's reaction time, changed depending on the exact nature of the task, and even reflected the subjects' errors. These responses arose at different times in different regions, allowing Tang et al. to suggest how signals flow through the frontal cortex during cognitive control.

In the future it will be important to further understand how the regions of the frontal cortex identified by Tang et al. interact with each other and to establish their roles in cognitive control. These observations could then be used to produce a theoretical framework that describes how the brain adapts behavior to different circumstances.

signals to bias processing in favor of the task-relevant stimuli (*Botvinick et al., 2001*; *Miller and Cohen, 2001*). The medial frontal cortex (mFC) also participates in cognitive control, possibly in a conflict monitoring capacity (*Botvinick et al., 2001*; *Ridderinkhof et al., 2004*; *Rushworth et al., 2004*). Recordings and lesions studies in the macaque anterior cingulate cortex (ACC) (*Ito et al., 2003*; *Nakamura et al., 2005*) suggest that ACC neurons are principally involved in monitoring for errors and making between-trial adjustments (*Brown and Braver, 2005*; *Ito et al., 2003*; *Johnston et al., 2007*; *Rothé et al., 2011*)—an idea that has received support by a recent study in the human ACC (*Sheth et al., 2012*). Recent work has also demonstrated that the supplementary motor area and the medial frontal cortex play an important role in monitoring for errors (*Bonini et al., 2014*). An alternative and influential theoretical framework posits that the ACC monitors for potential conflicts and subsequently directs the dlPFC to engage control processes (*Botvinick et al., 2001*; *Shenhav et al., 2013*). Several human neuroimaging studies are consistent with this notion (*Botvinick et al., 1999*; *Kerns, 2006*; *Kerns et al., 2004*; *MacDonald, 2000*) but the relative contributions of dlPFC, mFC, and ACC to cognitive control remain a matter of debate (*Aarts et al., 2008*; *Cole et al., 2009*; *Fellows and Farah, 2005*; *Mansouri et al., 2007*; *Milham and Banich, 2005*; *Milham et al., 2003*; *Rushworth et al., 2004*).

Previously, some neuroimaging studies have suggested that these frontal cortex regions can be differentiated based on the presence or absence of conflict signals (*MacDonald, 2000*). The challenge in dissociating the relative roles of these regions during Stroop-like tasks is that increased task difficulty recruits a host of executive functions (attention, decision-making, uncertainty, cognitive control). These functions are associated with neural activity spanning tens to hundreds of milliseconds and the underlying dynamics are difficult to untangle with the low temporal resolution of existing neuroimaging techniques (*Shenhav et al., 2013*). Human single neuron studies provide millisecond resolution but have focused on individual regions (*Sheth et al., 2012*). We took advantage of the high spatiotemporal resolution of intracranial recordings in human epilepsy patients and

the ability to record simultaneously from multiple regions to directly investigate the dynamics of conflict responses during cognitive control. We hypothesized that subregions of frontal cortex could be differentiated based on the temporal profile of their conflict responses. We recorded intracranial field potentials from 1397 electrodes in 15 subjects while they performed the Stroop task or a variation in which they were asked to read the word instead of focusing on its color.

We observed conflict-selective activity throughout several regions in frontal cortex: ACC, mFC, dlPFC, and also orbitofrontal cortex (OFC). Several analyses link these signals to cognitive control. Neural responses increased for incongruent compared to congruent trials, and these signals correlated with behavioral reaction time, depended on the task, and exhibited adaptation over trials. We compared pairs of simultaneously recorded electrodes to disassociate these different regions based on the timing of these conflict responses rather than their presence or absence. Conflict responses emerged first in the ACC and subsequently emerged in dlPFC and mFC and finally in OFC. These observations propose a plausible flow of signals underlying cognitive control.

## Results

We recorded field potentials from 15 epilepsy patients implanted with intracranial electrodes in frontal cortex as they performed the Stroop task (*Figure 1*, *Supplementary file 1*). After 500 ms of a fixation cross, subjects were presented with one of three words (Red, Blue, Green), which were colored

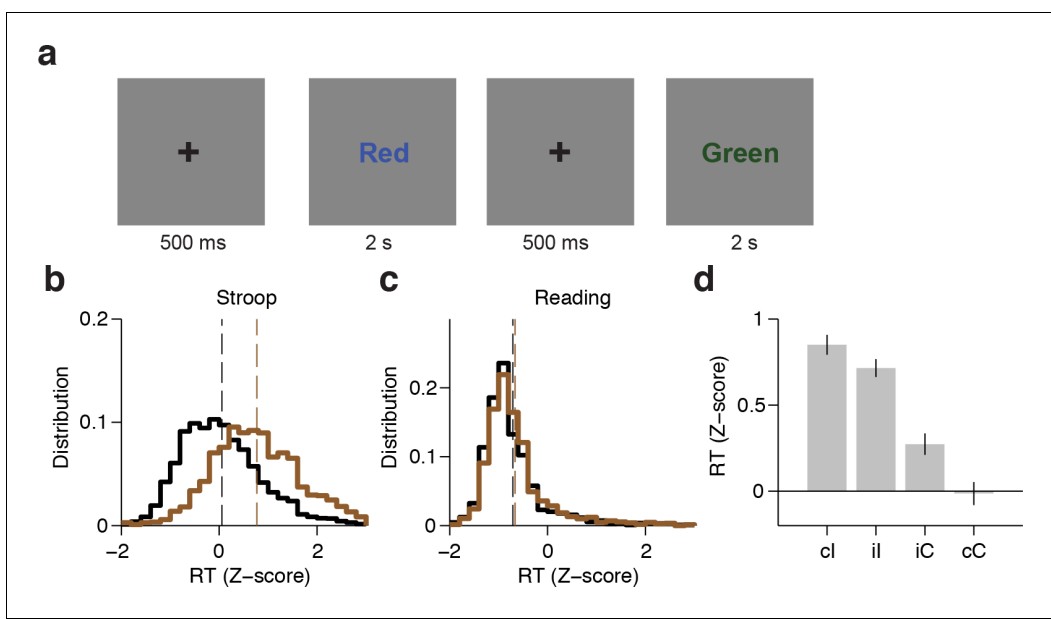

**Figure 1.** Experimental task and behavioral performance. (**A**) Subjects were presented with one of three words (Red, Blue or Green); each word was randomly colored red, blue, or green. Trials were incongruent (I) when the word and color did not match, and were congruent (C) otherwise. The word-color combinations were counterbalanced and randomly interleaved. Subjects performed the Stroop task (name the color), and the Reading task (read the word) in separate blocks. (**B**) Distribution of z-scored behavioral reaction times (speech onset) across all subjects (n = 15) for congruent (black) or incongruent (brown) trials during the Stroop task. Bin size = 0.2. Dashed lines indicate average reaction times. (**C**) Distribution of z-scored reaction times during the Reading task. (**D**) Z-scored reaction time across subjects for different trial histories during the Stroop Task (cI: incongruent trial preceded by congruent trial; iI: incongruent trial preceded by incongruent trial; iC: congruent trial preceded by incongruent trial; cC: congruent trial preceded by congruent trial). Error bars indicate s.e.m.

The following source data and figure supplement are available for figure 1:

**Source data 1.** Behavioral data.

**Figure supplement 1.** Behavioral data for each subject.

either, red, blue, or green. We refer to congruent trials (C) where the font color matched the word (60% of the trials) compared to incongruent trials (I) where the font color conflicted with the word (40% of the trials). Within each trial type, the word-color combinations were counter-balanced and randomly interleaved. The stimuli were presented for 2 s (in two subjects, for 3 s). Subjects were asked to respond verbally and either name the color (Stroop task), or read the word (Reading task) in separate blocks. Performance during congruent trials was essentially at ceiling (*Figure 1—figure supplement 1*).

An ANOVA conducted on subjects' performance with stimulus type (congruent or incongruent) and task (Stroop or Reading) as repeated measures revealed a significant interaction between stimulus type and task (F = 22.9, *P* < 0.001). For the Stroop task, subjects made more errors during incongruent trials (average error rate: 5 ± 3%, *P* < 0.001 paired t-test), as demonstrated in previous studies (*Bugg et al., 2008*; *Egner and Hirsch, 2005b*; *Kerns et al., 2004*). There was no difference in the number of error trials during the Reading task (*P* = 0.76, paired t-test). Subsequent analyses focused on correct trials only unless otherwise stated. Subjects' reaction times also had a significant interaction between stimulus type and task (F = 65.2, *P* < $10^{-5}$, ANOVA). Consistent with previous observations (*Stroop, 1935*), subjects' response times during the Stroop task were delayed for incongruent trials compared to congruent trials (*Figure 1B*, average delay: 215 ± 93 ms, *P* < 0.001, paired t-test, see also *Figure 1—figure supplement 1C* for individual subject data). The reaction time delays were shorter in the Reading task (*Figure 1C*, average delay: 22 ± 31 ms, *P* = 0.02, paired t-test). Trial history also has a strong effect on reaction time (known as Gratton effect in the literature [*Gratton et al., 1992*]). A repeated measures ANOVA revealed an interaction between previous and current trial type (F = 19.5, *P* < 0.001). Incongruent trials that were preceded by a congruent trial (cI trials) elicited slower reaction times compared to incongruent trials that were preceded by an incongruent trial (iI trials) (*Figure 1D*, average reaction time difference: 34 ± 14 ms, *P* = 0.03, paired t-test). A similar Gratton effect was found for iC versus cC trials (*Figure 1D*, average reaction time difference: 72 ± 136ms, *P* < 0.001, paired t-test).

We recorded intracranial field potentials from 1397 electrodes (average 93 ± 31 electrodes per subject) while subjects performed the Stroop and Reading tasks. The number of electrodes per subject and the location of these electrodes were strictly dictated by clinical needs. Therefore, there was a wide distribution of electrode locations, as is typical in this type of recordings (*Liu et al., 2009*). We excluded electrodes in epileptogenic regions. We focused on the neural signals in the gamma band (70–120 Hz) given their prominence in sensory, motor and cognitive phenomena (*Crone et al., 1998a*; *Liu et al., 2009*; *Oehrn et al., 2014*); results for other frequency bands are shown in *Figure 2—figure supplement 2* and *Figure 4—figure supplement 1* and *2*. Presentation of the visual stimuli evoked rapid and color/word selective neural responses in visual cortical areas within 200 ms of stimulus onset, as expected from previous studies (e.g. *Liu et al., 2009*). Other electrodes were activated for different motor (verbal) outputs (e.g. *Bouchard et al., 2013*; *Crone et al., 1998a*).

## Conflict responses in frontal cortex

We focused on 469 electrodes located in areas within frontal lobe which have been previously implicated in executive function: medial frontal cortex (mFC, n = 111), orbitofrontal cortex (OFC, n = 156), dorsolateral prefrontal cortex (dlPFC, n = 168) and the anterior cingulate cortex (ACC, n = 34). We applied a non-parametric analysis of variance (ANOVA) to measure whether and when the physiological responses differed between congruent and incongruent trials. An electrode was considered conflict-selective if the F-statistic was greater than a significance threshold computed by a permutation test with *P* = 0.001 for 50 consecutive milliseconds (Materials and methods). The latency was defined as the first time of this threshold-crossing.

*Figure 2* shows an example electrode from the left Anterior Cingulate Cortex that responded differentially between congruent and incongruent trials during the Stroop task. These signals were better aligned to the speech onset than to the stimulus onset, as shown in the response-aligned view (compare *Figure 2A–C* with *Figure 2D–F*). During the Stroop task, the response-aligned signals were significantly stronger for the incongruent (brown) trials compared to the congruent (black) trials (*Figure 2D*, *P* < $10^{-5}$, ANOVA), and were invariant to the particular word/color combinations (*Figure 2G*). Incongruent trials could be discriminated from congruent trials at a latency of 669 ± 31 ms (mean ± s.e.m.) before the onset of the response (*Figure 2D*). This conflict response was also

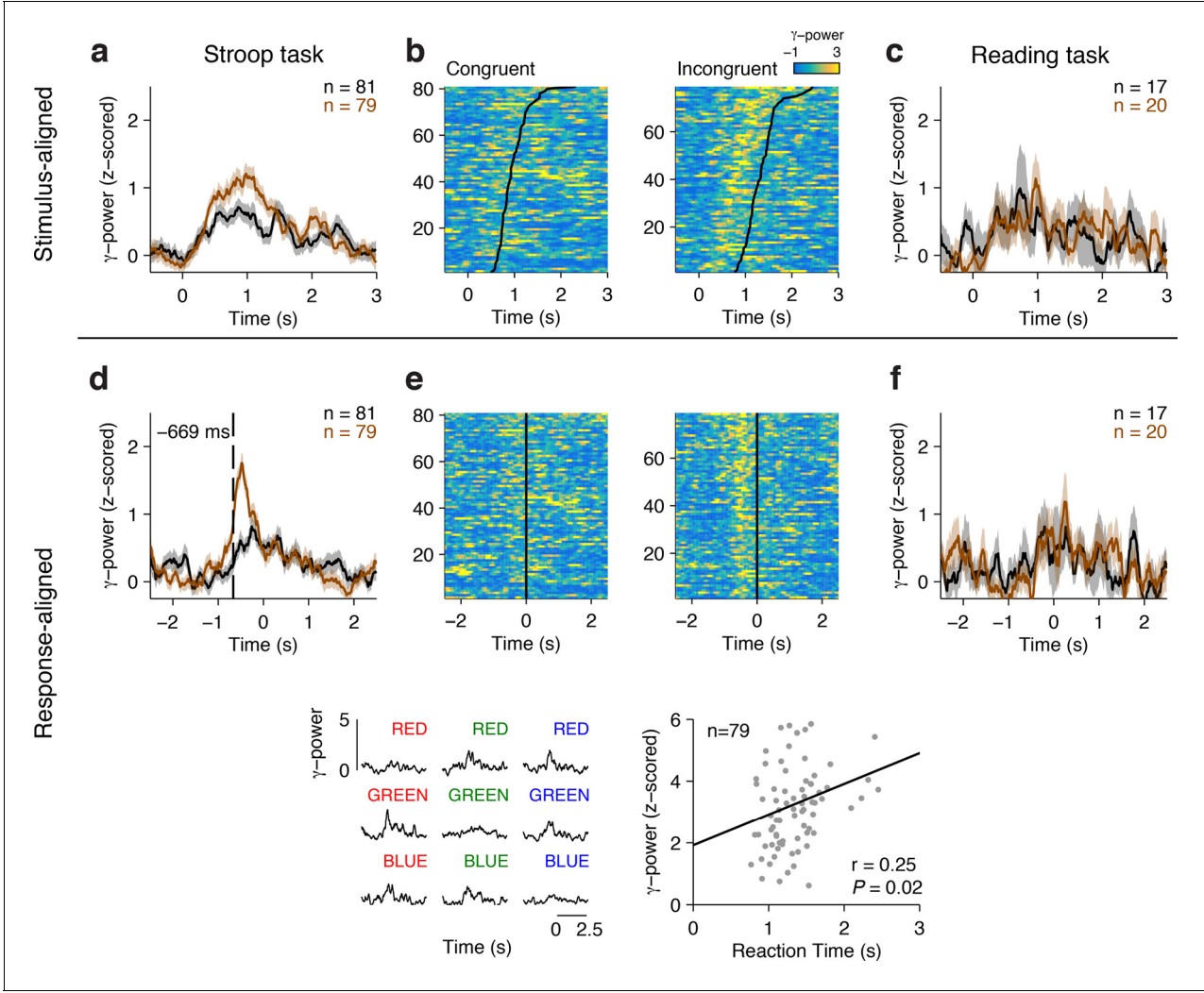

**Figure 2.** Example electrode in left Anterior Cingulate Cortex. (A) Average gamma power signals aligned to the stimulus onset from an electrode during the Stroop task, for congruent (black) or incongruent (brown) stimuli. For display purposes only, we z-scored the gamma power by subtracting the average and dividing by the standard deviation of power during the baseline period (500 ms prior to stimulus onset). Shaded areas indicate s.e.m. The total number of trials for each condition is indicated in the upper right. (B) Single-trial data for congruent (left) and incongruent (right) trials. Each row is a trial, and the color indicates the z-scored gamma power (color scale on upper right). Trials are sorted by behavioral response time (black line). (C) Same as (A), but showing data from the Reading task. (D-F) Same as in A-C, but aligning the data to behavioral response time. Gamma power was better aligned to the behavioral response, and was stronger for incongruent compared to congruent trials. The dashed line indicates the response-aligned latency, defined as the first time point at which incongruent and congruent trials can be discriminated. (G) Signals elicited by each of the 9 possible stimulus combinations. (H) There was a correlation between the maximal z-scored gamma power and behavioral reaction times during incongruent trials (Pearson correlation coefficient = 0.25, P = 0.02, permutation test). Each point in this plot represents a single trial.

The following source data and figure supplements are available for figure 2:

**Source data 1.** Conflict-selective electrode data.

**Figure supplement 1.** Example conflict-selective electrode in the right dorsolateral Prefrontal Cortex.

**Figure supplement 2.** Example conflict-selective electrode in the Orbitofrontal Cortex comparing responses in the Theta and Gamma Bands.

specific to the Stroop task; there was a significant interaction between congruency and task (F = 13.5, P = 0.007, ANOVA). The same stimuli did not elicit differential activity during the Reading task (*Figure 2F*). We assessed the correlation between the neural signal strength and behavioral reaction

times in single trials. The maximal gamma power during each incongruent trial (using the average gamma power yielded similar results) was positively correlated with the behavioral reaction times (*Figure 2H*, ρ = 0.25, *P* = 0.02).

Any differences between congruent and incongruent trials in the stimulus-aligned analyses can be confounded by the reaction time differences; therefore, we focus subsequent analyses on the response-aligned signals. More example electrodes are shown in *Figure 2—figure supplement 1* (dlPFC) and *Figure 2—figure supplement 2* (OFC).

Using the aforementioned criteria, we identified n = 51 conflict selective frontal cortex electrodes during the Stroop task, with contributions from 13 subjects (*Supplementary files 2* and *3*). These electrodes were distributed throughout different subregions within frontal cortex (*Figure 3A*). To evaluate whether random variation in the signals could give rise to apparent conflict-selective electrodes, we randomly shuffled the congruent/incongruent trial labels 10,000 times and applied the same statistical criteria (Materials and methods). Across our population, we found n = 4.4 ± 0.03 false positive electrodes (mean ± s.e.m., out of 469 electrodes), which corresponds to a false discovery rate (FDR) of q = 0.01, which is significantly less than our observation of n = 51 electrodes. The number of conflict-selective electrodes within each subregion was significantly greater than expected by chance (*Figure 3B*, *P* < 0.01, all regions). We repeated the analyses during the Reading task. In contrast with the Stroop task, we only observed n = 3 conflict-selective frontal cortex electrodes during the Reading task (out of 469 electrodes), a number that is within the false positive rate.

To account for within-subject and across-subject variation, we used a multilevel model (*Aarts et al., 2014*) to conduct a group analysis of the physiological responses, with electrodes nested within subjects (Materials and methods). Across the population, we observed a significant interaction between the factors congruency and task on the gamma power ($\chi^2$=9.2, *P* = 0.002). Consistent with the single electrode examples, gamma power was greater for incongruent compared to congruent trials, but only during the Stroop task (*Figure 4A*, Stroop: *P* < $10^{-3}$, Reading: *P* = 0.56). We computed the average response in each region (*Figure 4B*). Each electrode's response was normalized by dividing the power during incongruent trials by the power in congruent trials (dividing the brown curve by the black curve in *Figure 2*), computing the logarithm and finally pooling within each region. The pooled responses in the OFC are visually less compelling (*Figure 4B*, bottom right subplot) due to the heterogeneity in the latency of the individual electrodes but the responses in the OFC were as vigorous as the ones in other areas (e.g. *Figure 2—figure supplement 2*). Similar conclusions were reached when plotting the pooled responses aligned to stimulus onset (*Figure 4—figure supplement 3*).

## Behavioral relevance of physiological responses

Several lines of evidence demonstrate a link between the neural signals described in the previous section and cognitive control: the neural signals correlated with reaction times, showed behavioral adaptation, and demonstrated error monitoring.

As shown in previous studies, there was a wide distribution of behavioral reaction times (*Figure 1B*). Consistent with the example electrode in *Figure 2*, behavioral reaction times across the population correlated with the strength of the physiological signals, even after controlling for trial history (*Figure 4C*, *P* < $10^{-5}$, sign-rank test).

The strength of these neural signals also revealed a neural correlate of the behavioral Gratton effect documented in *Figure 1D*: gamma power was greater in cI compared to iI trials (*Figure 4D*). Using the aforementioned multilevel model, we found a significant interaction between trial history (cI or iI) and task ($\chi^2$=4.4, *P* = 0.03). This Gratton effect was stronger in the Stroop task (*P* < 0.001) than in the Reading task (*P* = 0.72). These differences were not observed for cC versus iC trials, where the interaction was not significant ($\chi^2$=1.9, *P* = 0.17) (*Figure 4E*). This analysis was performed after removing stimulus repetition trials. The Gratton effect was present in all four frontal regions and there were no statistically significant differences in the strength of the effect across regions (F = 0.25, *P* = 0.86, ANOVA). To control for reaction time effects on these comparisons, we ran an analysis of covariance (ANCOVA) to test for a main effect of trial history on the gamma power with the behavioral reaction time as a covariate (Materials and methods). The neural Gratton effect during the Stroop task persisted under these controlled conditions (*P* = 0.0002, multilevel model). We also explicitly ruled out reaction time differences by subsampling to match the reaction time distribution between conditions, with similar results (*P* = 0.01, multilevel model). Together, these results suggest

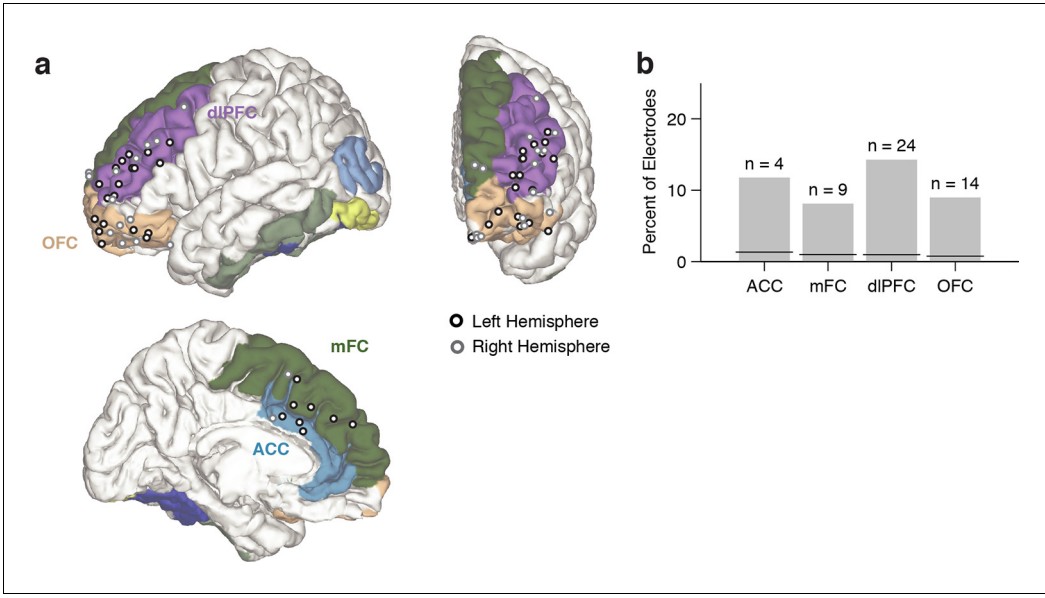

**Figure 3.** Electrode locations. (**A**) Location of conflict-selective electrodes (black/gray) shown on a reference brain, with each region colored (Materials and methods). Electrodes from the right hemisphere were mapped to the left hemisphere for display purposes. For more detail, see *Supplementary file 2*. (**B**) Percent of total electrodes in each region that were selective for conflict. Chance levels were computed using a permutation test (black line). The number of observed electrodes was significantly above chance for all regions ($P < 0.01$, permutation test, Materials and methods).

The following source data is available for figure 3:

**Source data 1.** Population gamma-power data.

that the neural signals described here code for an internally perceived level of conflict that exhibits conflict adaptation and correlates with the across-trial variability in reaction times.

## Conflict responses in other frequency bands

The results presented above focus on the neural signals filtered within the gamma frequency band (70–120 Hz). We also examined the responses elicited in the broadband signals (1 to 100 Hz) as well as in the theta, (4 to 8 Hz), beta, (9 to 30 Hz), and low gamma (30–70 Hz) bands. No conflict selective responses were observed in the broadband signals or low gamma band. We found conflict-selective responses both in the theta and beta bands (see example in *Figure 2—figure supplement 2a–f*). Across theta and beta frequency bands, we also observed a significant interaction between Congruency and Task (theta: $P < 10^{-5}$, beta: $P < 10^{-4}$, multilevel model). Consistent with the results reported in the gamma frequency band, conflict responses in the theta and beta bands were more prominent during the Stroop task compared to the Reading task (*Figure 4—figure supplement 1*). In contrast to the results in the gamma band, power in the theta and beta bands *decreased* during incongruent trials. Furthermore, power in the theta and beta frequency bands was not correlated with reaction times (theta: $P = 0.43$, beta: $P = 0.09$, sign-rank test).

In addition to separately examining the responses in different frequency bands, an important aspect of encoding of cognitive information is the relationship between signals across frequencies. In particular, several studies have demonstrated that the amplitude of the gamma band is coupled to the phase of slower oscillations in the theta band (*Canolty et al., 2006*; *Oehrn et al., 2014*; *Tort et al., 2008*). We therefore examined the degree of cross-frequency coupling between the signals in the gamma and theta bands (*Figure 4—figure supplement 2*). Consistent with previous studies, we found that 50% of the electrodes demonstrated significant theta-gamma coupling. However, the strength of this coupling was not different between congruent and incongruent trials across the population of conflict-selective electrodes ($P = 0.52$, sign-rank test).

## Error monitoring signals

The conflict responses reported above are based on correct trials only. Yet, error monitoring has also been ascribed to frontal cortical circuits (*Bonini et al., 2014*; *Shenhav et al., 2013*; *Yeung et al., 2004*). To investigate whether the same electrodes responding to conflict are also involved in successful error monitoring, we analyzed the neural signals during self-corrected trials. In these trials, subjects initially made an erroneous response and rapidly corrected themselves with the

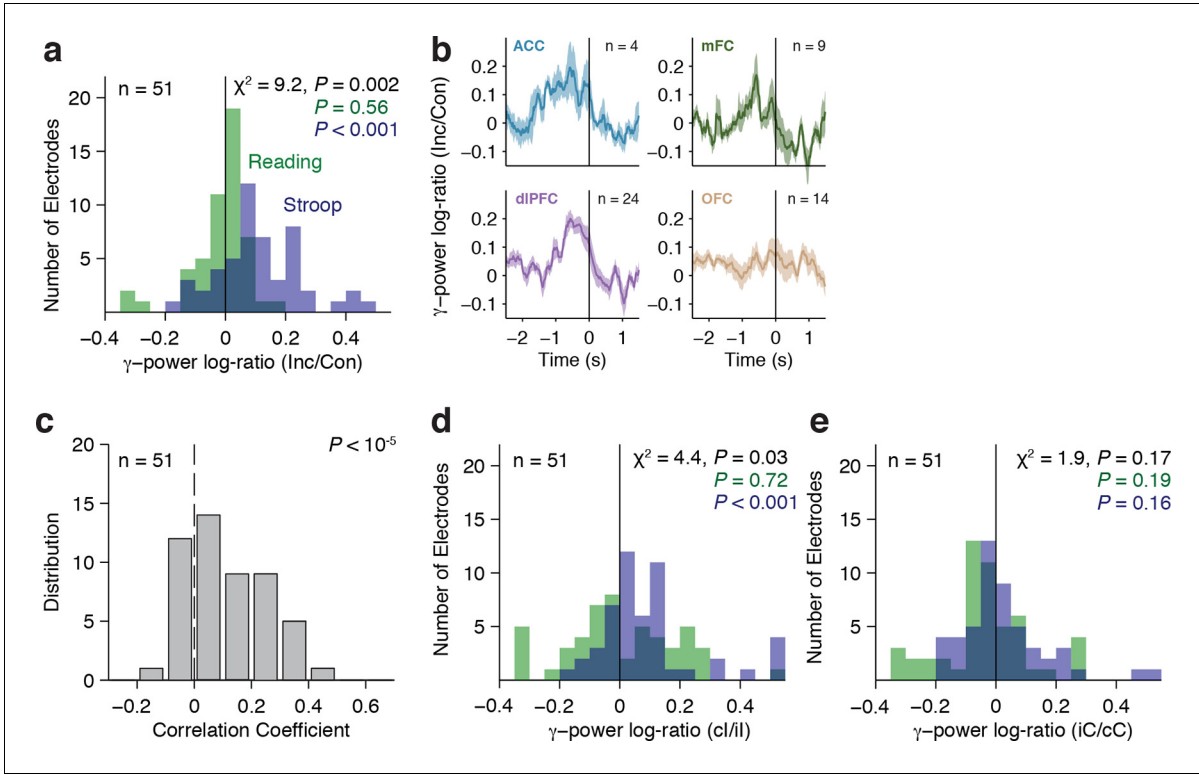

**Figure 4.** Gamma power in frontal cortex correlates with behavior. (**A**) Distribution of gamma power log-ratio (Incongruent/Congruent) for the Stroop task (blue) and Reading task (green). Bin size = 0.05. Gamma power showed a significant interaction between Congruency and Task (*P* = 0.002, multilevel model, Materials and methods). Power was larger for incongruent versus congruent trials during the Stroop task (*P* < 0.001, n = 51 frontal cortex electrodes) but not during the Reading task (green, *P* = 0.56). The statistical analyses directly compare the gamma power, we show the log-ratios here for display purposes only. (**B**) Normalized gamma power log-ratio averaged across electrodes from each of the four different frontal cortex regions during the Stroop task. We divided the power during incongruent trials by the power during congruent trials, then computed the log and finally averaged across electrodes. Data are aligned to the behavioral response onset (t=0). (**C**) Distribution of Pearson correlation coefficients between the maximal gamma power and behavioral reaction time during incongruent trials for n = 51 frontal cortex electrodes. These correlations were significantly positive (P < $10^{-5}$, sign-rank test). Bin size = 0.1. (**D**) For incongruent trials, there was a significant interaction between trial history and task (*P* = 0.03, multilevel model). Gamma power was larger for incongruent trials preceded by congruent trials (cI) compared to incongruent trials preceded by incongruent trials (iI), particularly during the Stroop task (blue, *P* = 0.001), compared to the Reading task (green, *P* = 0.72). Data beyond the range of the x-axis are shown in the first or last bins. (**E**) For congruent trials, there was no interaction between trial history and task (*P* = 0.17, multilevel model). Gamma power was similar in congruent trials preceded by incongruent trials (iC) compared to congruent trials preceded by congruent trials (cC) during the Stroop task (blue, *P* = 0.16) and during the Reading task (green, *P* = 0.19).

The following source data and figure supplements are available for figure 4:

**Source data 1.** Population gamma-power data.

**Figure supplement 1.** Theta and Beta band population results.

**Figure supplement 2.** Cross-frequency coupling analyses.

**Figure supplement 3.** Stimulus-aligned population averages.

right answer. Given the high performance level of all subjects, the number of such trials is low. However, these trials are particularly interesting because we can be certain that there was successful error detection (as opposed to error trials without any self-correction). An example self-corrected trial from the ACC electrode shown previously is illustrated in *Figure 5A*. The subject initially made an incorrect response (green), which was rapidly followed with the correct response (red). Increased gamma power was observed after onset of the erroneous response. In contrast, the following corrected behavioral response exhibited no such post-response signal. Additionally, these error-monitoring signals were not observed in correct incongruent trials (*Figure 2D*), and were consistent across the n = 11 self-corrected trials for this subject (*Figure 5B*, P = 0.001, signed rank test). Another example electrode is shown in *Figure 5C–D*. There were only two subjects contributing n = 7 conflict-signaling electrodes that had a sufficient number of self-correction trials (greater than five trials) for this analysis. For each electrode, we compared the difference in neural signals during the one-second post-response window between the initial error and the following self-correction. Of those n = 7 electrodes, n = 5 electrodes showed evidence of error monitoring (*Figure 5E*, P < 0.05, sign-rank test). Although the number of electrodes and trials in this analysis is small, these results provide a direct correlate of error monitoring signals. Furthermore, these results highlight that the same electrodes that respond to conflict leading up to the behavioral response can also show post-response error monitoring.

## Regional differences in conflict response latencies

We observed conflict-selective responses in the anterior cingulate cortex, medial frontal cortex, dorsolateral prefrontal cortex and orbitofrontal cortex. To examine the dynamics of cognitive control orchestrating the transformation of conflicting visual signals to motor outputs, we compared, across those four regional groups, the latencies relative to behavioral response onset at which the congruent and incongruent trials could be discriminated. Comparing latencies across regions is difficult, especially across subjects with varying reaction times. For a controlled and direct comparison, we restricted the analysis to compute the latency differences between pairs of simultaneously recorded electrodes. This within-subject pairwise analysis had increased power to examine the relative dynamics between frontal lobe areas (*Figure 6*). The relative latencies were significantly different across the regions (P = 0.01, permutation test, post-hoc testing was controlled for multiple comparisons using the Benjamin-Hochberg procedure, Materials and methods). Conflict responses in the ACC preceded those in all the other frontal lobe regions, followed 207 ± 40 ms later by dorsolateral prefrontal cortex and 388 ± 83 ms later by medial frontal cortex. Signals in orbitofrontal cortex emerged 319 ± 78 ms after dlPFC. This entire processing cascade took approximately 500 ms. For comparison, subjects' behavioral reaction times to incongruent trials were 1105 ± 49 ms. The latency difference between ACC and dlPFC is based on 6 electrode pairs: one ACC electrode and six simultaneously recorded dlPFC electrodes. There was only one pair of simultaneous recordings between ACC and OFC and we do not report this value in *Figure 6*. The other region comparisons have contributions from multiple electrodes in multiple subjects (*Supplementary file 3*). These results suggest a temporal hierarchy of cognitive control mechanisms culminating in speech onset.

## Discussion

We used intracranial field potentials to measure the dynamics of conflict responses across frontal cortex leading up to the behavioral response in the Stroop task. Previous physiological and functional neuroimaging studies have documented the involvement of multiple of these frontal cortex areas in the Stroop or similar tasks (*Botvinick et al., 1999*; *Kolling et al., 2012*; *MacDonald, 2000*; *Niendam et al., 2012*; *Oehrn et al., 2014*; *Sheth et al., 2012*). The intracranial field potential recordings reported here show conflict-selective signals in ACC (e.g. *Figure 2*), dlPFC (e.g. *Figure 2—figure supplement 1*), mFC (e.g. *Figure 4B*) and OFC (e.g. *Figure 2—figure supplement 2*). The mFC and dlPFC have been previously implicated in cognitive control, and these structures are extensively connected to the rest of frontal cortex areas (*Ridderinkhof et al., 2004*). The role of the OFC in cognitive control during Stroop-like tasks has not been reported previously, possibly because of technical challenges in neuroimaging near this area (*Weiskopf et al., 2006*).

We presented several lines of evidence that demonstrate that these conflict-selective physiological signals are relevant for behavior during the Stroop task. Longer behavioral reaction times were

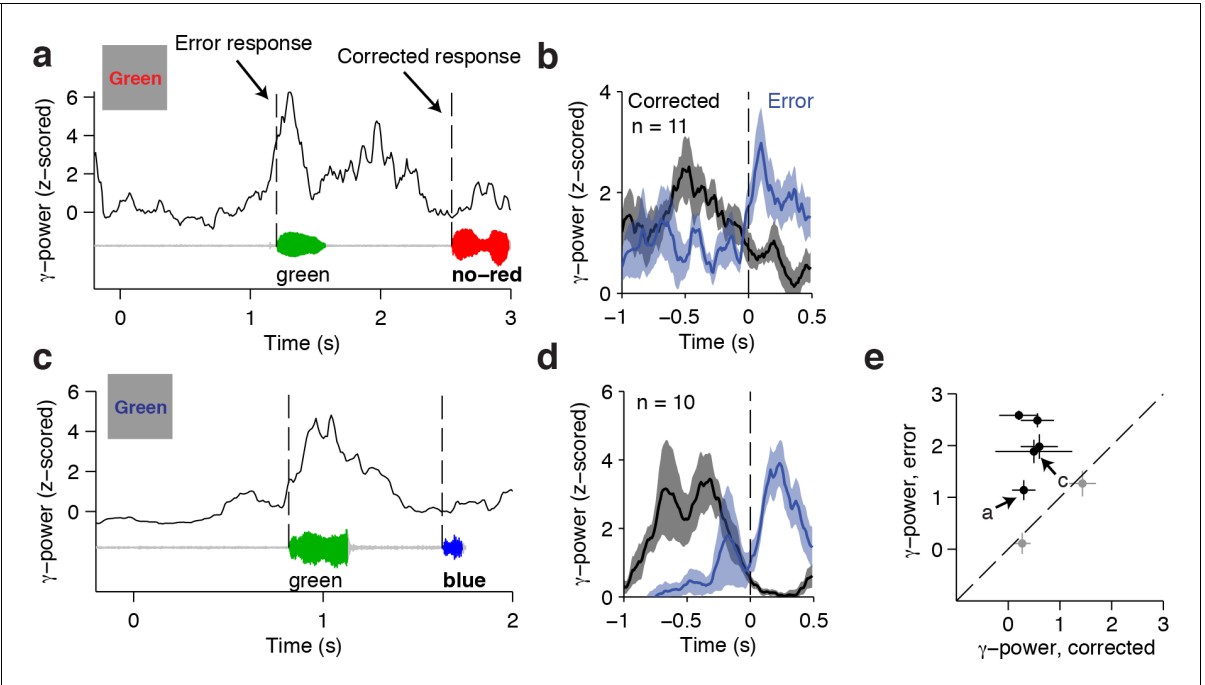

**Figure 5.** Responses during self-corrected error trials. (A) An example self-correction trial from the ACC electrode in *Figure 2* when the word Green colored in red was presented. The single trial gamma power is shown on top, with the speech waveform below. The dashed lines indicate the onset of the initially incorrect response ('green') and the following corrected response in bold ('no – red'). Note the increased gamma power after an error response. (B) Average gamma power aligned to the onset of the initial error response (blue) and the onset of the corrected response (black) for n = 11 self-correction trials. Shaded areas indicate s.e.m. The post-response power was significantly greater after the error (P = 0.001, signed-rank test). (C-D) Same as (A-B) for another example electrode in the dorsolateral prefrontal cortex. The post-response power was significantly greater after the error response (P = 0.002, signed-rank test). (E) Across the n = 7 electrodes with n = 10 or greater self-correction trials, the z-scored gamma power during the initial error response was larger than during the corrected response. Electrodes with significant differences (P < 0.05, signed-rank test) are colored black. Letters mark the examples in (A) and (C).

The following source data is available for figure 5:

**Source data 1.** Data for self-correcting trials.

correlated with greater gamma power on a trial-by-trial basis during the Stroop task but not during the Reading task, even after accounting for trial history and for differences between congruent and incongruent stimuli (*Figure 2H*, *4C*). The same identical stimuli can elicit a range of behavioral reaction times and this internal degree of conflict can be captured, at least partly, by the strength of gamma power in frontal cortex in each trial.

The neural correlates of behavioral adaptation (Gratton effect [*Gratton et al., 1992*]) were observed in the ACC, consistent with prior studies based on human single neuron recordings (*Sheth et al., 2012*), neuroimaging (*Botvinick et al., 1999*; *Kerns, 2006*) and also in accordance with the behavioral effects of ACC resection (*Sheth et al., 2012*). Conflict responses throughout the other frontal cortex regions also demonstrated the neural Gratton effect, suggesting a more distributed network involved in across-trial adaptation than previously hypothesized. The physiological responses in these areas were stronger in cI trials (incongruent trials that were preceded by congruent trials) than iI trials (*Figure 4D*). While the increased activity in cI trials compared to iI trials is consistent with neuroimaging studies (*Botvinick et al., 1999*), single neuron recordings in a different Stroop-like task report the opposite relationship (iI > cI) (*Sheth et al., 2012*). These differences point to potentially interesting distinctions between the activity of individual neurons and coarser population measures that warrant further investigation.

Another discrepancy between neuroimaging studies and single unit recordings is the presence of conflict responses and error signals. Single unit recording in macaque ACC typically find error monitoring signals but not conflict-selective responses (*Cole et al., 2009*; *Emeric et al., 2010*; *Ito et al.,*

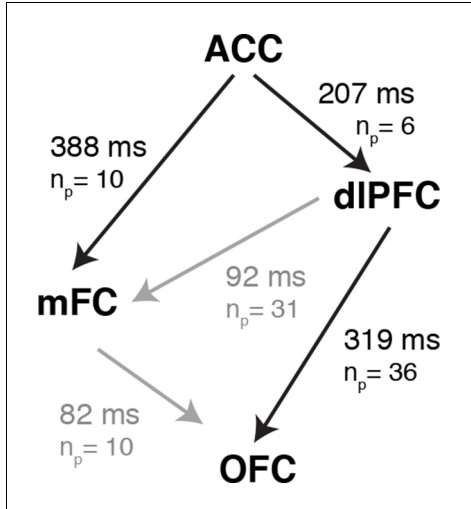

**Figure 6.** Latency Comparisons across regions. Latency differences between different regions computed from all pairs of simultaneously recorded electrodes. $n_P$ denotes the number of electrode pairs. Because we only consider simultaneously recorded electrodes here, not all the electrodes modulated by conflict can be paired with any other electrode. *Supplementary file 3* shows the number of electrodes modulated by conflict in each area and subject. There was only one electrode pair between ACC and OFC and therefore we do not show the latency difference between these two regions here. Significant latency differences ($P < 0.05$, permutation test, Materials and methods) are shown in black, and non-significant differences in gray. ACC leads both mFC ($P = 0.001$) and dlPFC ($P = 0.02$), with OFC following dlPFC ($P = 0.009$).

The following source data is available for figure 6:

**Source data 1.** Data for region latency comparisons.

2003; *Taylor et al., 2006*), see however (*Ebitz and Platt, 2015*), whereas human neuro-imaging studies report both types of signals in ACC. There has been significant debate concerning whether action monitoring and conflict detection represent distinct processes (*Carter et al., 1998*; *Carter et al., 2000*; *Nee et al., 2011*; *Swick and Turken, 2002*). Because both processes may co-occur on the same trials, high temporal resolution is required to disassociate the two computations. A recent human intracranial study has found error signals in supplementary motor area and medial frontal cortex (*Bonini et al., 2014*), and a human single unit study reported conflict signals in ACC (*Sheth et al., 2012*). The current work demonstrates the coexistence of both error signals and conflict signals. The analysis of the few self-correction trials in our data suggests that the same areas responsible for pre-behavioral conflict signals can also produce post-behavioral response error-monitoring signals (*Figure 5*). In addition, the relative timing of the conflict and error signals surrounding the neural responses confirms computational predictions based on a connectionist architecture to explain the mechanisms of conflict (*Yeung et al., 2004*) and scalp EEG studies (*Hughes and Yeung, 2011*). These results are consistent with computational models suggesting that these signals may represent a general error-likelihood prediction, of which conflict and error detection are special cases (*Brown and Braver, 2005*).

It has been suggested that ACC and supplementary eye field neurons in macaque monkeys respond to specific stimulus and/or behavioral combinations but are not directly modulated by conflict (*Cole et al., 2009*; *Nakamura et al., 2005*). At the level of the intracranial field potentials reported here, the modulation of conflict trials observed in the four frontal cortex regions could not be ascribed to specific stimulus or behavioral responses (e.g. *Figure 2G*) and were also task dependent (compare *Figure 2A* versus *2C*). In these patients, we did not have access to single neuron responses and we therefore cannot rule out the possibility that individual neurons show distinct patterns of responses that are averaged out at the field potential level.

Besides the high gamma band, we also observed conflict responses in the beta and theta bands, but not the low gamma band (e.g. *Figure 4—figure supplement 1*). Previous work has suggested differential roles for distinct oscillatory components of the local field potential (*Cavanagh and Frank, 2014*; *Kahana et al., 2001*; *Ullsperger et al., 2014*; *von Stein and Sarnthein, 2000*). There were clear differences in the type of information conveyed by distinct frequencies components. Lack of significant correlations with reaction time in the theta and beta bands suggests that the gamma band better captures the behavior. Additionally, conflict responses were characterized by increased power in the gamma band, but decreased power in the theta and beta bands (*Figure 4—figure supplement 1*). Previous scalp EEG recordings (*Cavanagh and Frank, 2014*; *Ullsperger et al., 2014*; *van Driel et al., 2015*) have demonstrated that conflict and/or error trials elicit increased theta power, suggesting potentially interesting differences in how theta is captured across spatial scales. We also observed a decrease in beta power, which is consistent with previous studies that correlate

frontal cortex activation with desynchronization in the beta band and increased synchronization in the gamma bands (*Crone et al., 1998a*; *Crone et al., 1998b*). Differences across tasks, recording methods, and targeted regions should be interpreted with caution. The roles of different oscillatory components in neocortex are not clearly understood. One possibility is that lower frequency bands reflect the summed dendritic input of the nearby neural population (*Logothetis et al., 2001*; *Mitzdorf, 1987*) and can act as channels for communication (*Cavanagh and Frank, 2014*), whereas higher frequency bands represent the population spiking rate (*Buzsaki et al., 2012*; *Ray and Maunsell, 2011*). Along these lines, we speculate that the theta desynchronization we observe could reflect a reduction of inputs, leading to inhibition of the prepotent but erroneous response.

While we observed conflict responses throughout frontal cortex, the spatiotemporal resolution of our intracranial recordings allowed us to separate regions by the latency at which conflict-selective responses emerge with respect to speech onset. By comparing pairs of simultaneously recorded electrodes, we found that conflict responses in the ACC lead the dlPFC by ~200 ms. Medial frontal cortex is anatomically close and extensively connected to the ACC, and the two regions are often grouped together (*Cavanagh et al., 2009*; *Ridderinkhof et al., 2004*). Yet, conflict responses in the mFC trail the ACC by hundreds of milliseconds, suggesting an important distinction between the two regions (*Rushworth et al., 2004*). The relative latency measurements place the OFC at the bottom of this cascade. The hierarchical cascade of processes described here is consistent with predictions from mechanistic models of cognitive control (e.g. see *Figure 2* in *Shenhav et al, Neuron 2013*). In particular, stimulus related signals are evident along the ventral visual stream early on and feed onto frontal cortex, where we find that ACC activity precedes activity in other frontal regions, followed by dlPFC, and finally mFC, and OFC.

Since the local field potential pools over many neurons, latency measures can be influenced by a variety of factors, such as the proportion of neurons selective for conflict and their laminar organization. Yet, at least in the ACC, the temporal profile of conflict responses we observed is similar to responses from human single unit recordings (*Sheth et al., 2012*). The relatively long delays between regions are also particularly intriguing. There are monosynaptic connections that link these four regions within frontal cortex and yet, it takes 100–200 ms to detect the relative activation between these areas (*Figure 6*).

Daily decisions require integration of different goals, contexts, input signals, and the consequences of the resulting actions. The current study provides initial steps to elucidate not only which brain areas participate in cognitive control on a trial-by-trial basis but also their relative interactions and differential roles. The relative latency measurements and correlations between neural activity and reaction time provide a framework to constrain theories of cognitive control, and propose a plausible flow of conflict responses through frontal cortex.

## Materials and methods

### Subjects
Subjects were 15 patients (10 male, Ages 10–50, *Supplementary file 1*) with pharmacologically intractable epilepsy treated at Children's Hospital Boston (CHB), Johns Hopkins Medical Institution (JHMI), Brigham and Women's Hospital (BWH), or Taipei Veterans General Hospital (TVGH). These subjects were implanted with intracranial electrodes in frontal cortex for clinical purposes. Five other subjects participated in this task but they were excluded from the analyses because they did not have any electrodes in frontal cortex. All studies were approved by each hospital's institutional review boards and were carried out with the subjects' informed consent.

### Recordings
Subjects were implanted with 2 mm diameter intracranial subdural electrodes (Ad-Tech, Racine, WI, USA) that were arranged into grids or strips with 1 cm separation. Electrode locations were determined by clinical considerations. There were 1397 electrodes (15 subjects). Sampling rates ranged from 256 Hz to 1000 Hz depending on the equipment at each institution: CHB (XLTEK, Oakville, ON, Canada), BWH (Bio-Logic, Knoxville, TN, USA), JHMI (Nihon Kohden, Tokyo, Japan), and TVGH (Natus, San Carlos, CA). All the data were collected during periods without any seizure events or immediately following any seizures.

## Task procedures

A schematic of the task is shown in *Figure 1*. After 500 ms of fixation, subjects were presented with a word stimulus for 2 s. The stimulus presentation was 3 s in two subjects. Stimuli were one of three words (Red, Blue, Green) presented in the subjects' primary language (CHB, BWH, JHMI: English; TVGH: Mandarin) either in red, blue, or green font color. Stimuli subtended approximately 5 degrees of visual angle and were centered on the screen. Trials were either congruent (C), where the font color matched the word, or incongruent (I), where the font color conflicted with the word. The order of congruent and incongruent trials was randomized. Approximately 40% of the trials were incongruent trials. Within congruent trials and within incongruent trials all color-word combinations were counter balanced and randomly interleaved. Subjects were asked to either name the color (Stroop task) or read the word (Reading task) within the time limit imposed by the stimulus presentation time.

Each block contained 18 trials, and the two tasks were completed in separate blocks. Most subjects completed 18 blocks of the Stroop task and 9 blocks of the Reading task (*Supplementary file 1*). Audio was recorded using a microphone at 8192 Hz sampling rate. No correct/incorrect feedback was provided.

## Electrode localization

Electrodes were localized by co-registering the preoperative magnetic resonance imaging (MRI) with the postoperative computer tomography (CT) (*Destrieux et al., 2010*; *Liu et al., 2009*). In 4 subjects without a postoperative CT, electrodes were localized using intraoperative photographs and preoperative MRI. For each subject, the brain surface was reconstructed from the MRI and then assigned to one of 75 regions by Freesurfer. Depth electrodes were assigned to either a subcortical structure or to gyri/sulci.

We focused on those electrodes in four frontal cortex regions (ACC: anterior and middle-anterior cingulate gyrus, mFC: superior frontal gyrus, dlPFC: middle frontal gyrus, and OFC: orbitofrontal gyrus).

## Behavioral analyses

To determine the behavioral reaction time for each trial, the short-time energy was computed from the audio recordings. For an audio signal *x(t)*, the short-time energy *E*(*t*) is defined as:

$$E(t) = \sum_{m=0}^{m=T} [x(m)w(t-m)]^2,$$

where *T* is the length of the recording and *w(t)* is a 300-point Hamming window (~40 ms). Speech onset was defined as the first time when the energy crossed a threshold set as 1 standard deviation above the baseline. Only trials where the subject gave a single verbal response and the speech onset could be identified were considered correct trials.

## Preprocessing

Unless otherwise noted, analyses in this manuscript used correct trials only. Electrodes with significant spectral noise were excluded from analysis (n = 25 out of 1397 total electrodes). For each electrode, a notch filter was applied at 60 Hz, and the common average reference computed from all channels was subtracted. Power in the theta (4–8 Hz), beta (9–30 Hz), and high-gamma band (70–120 Hz) was extracted using a moving window multi-taper Fourier transform (Chronux toolbox [*Mitra and Bokil, 2008*]) with a time-bandwidth product of five and seven tapers. The window size was 200 ms with 10 ms increments. In several figures, the gamma power was z-scored for display purposes (see figure legends).

## Analyses of neural response selectivity

To determine whether and when an electrode responded selectively to conflict, we used a sliding F-statistic procedure (*Liu et al., 2009*). Electrodes with differential responses between congruent and incongruent trials were selected by computing the F-statistic, for each time bin, comparing the neural responses between congruent and incongruent trials. Electrodes were denoted as 'conflict selective' if (1) the F-statistic exceeded a significance threshold for 50 consecutive milliseconds, and (2)

the average neural response exceeded one standard deviation above the baseline period at least once during the trial. A null distribution generated by randomly permuting the labels was used to set the significance threshold with $P = 0.001$. The latency at which congruent and incongruent stimuli could be discriminated was defined as the first time of this threshold crossing. For the response-aligned view, only electrodes where the latency preceded the response were included in subsequent analysis. This selection process was independently performed for each electrode in both stimulus-aligned and response-aligned analyses, and separately for the Stroop and Reading task.

We used a permutation test with 10,000 shuffles to obtain a false discovery rate for our selection process. The congruent/incongruent trial labels were randomized 10,000 times and we measured the average number of electrodes across our population that passed the selection procedure.

## Single electrode analyses

For the selected electrodes obtained with the procedure described in the previous section, we performed a number of within-electrode analyses. We measured single-trial correlations with behavioral reaction times, assessed the significance of interactions and simple/main effects, and controlled for confounds in measuring the neural Gratton effect.

### Single-trial analyses

For single trial comparisons across conditions, signal power for each trial was computed for both response-aligned and stimulus-aligned analyses. For stimulus-aligned data, the signal power was defined as the maximal power from stimulus onset to 1 s after stimulus onset. For response-aligned analyses, the signal power was defined as the maximal power from one second before the response to the response onset. Analyses using the average power within the same window yielded similar results. Single-trial response latency was defined as the time of maximal activation relative to stimulus onset.

### Interaction effects

For conflict-selective electrodes, we measured the significance of task dependence by performing, at each time bin, an ANOVA on the gamma power with the factors Congruency and Task (*Nieuwenhuis et al., 2011*). The peak F-statistic of the interaction term over the pre-response window was compared against a null distribution generated by randomly shuffling the trial labels. Simple effects were tested using this same approach.

### Neural Gratton effect

We evaluated the neural signal difference between trials with different histories (e.g. cI versus iI), while removing trials with stimulus repetitions. Given that (1) reaction times are different for the cI versus iI trials (*Figure 1*) and (2) gamma power is significantly correlated with reaction time in incongruent trials (*Figure 4*), we would expect differences in gamma power in cI versus iI trials. To control for this potential confounding effect in our measurements of trial history dependence, we applied two methods. First, for each electrode, we performed an ANCOVA on the gamma power with trial history (cI or iI, for example) as the group and reaction time as a covariate. We computed the regression line, extracted the RT-adjusted gamma power from the y-intercept and used this value in the group analysis. Second, we performed a matched reaction time analysis, where the distribution of reaction times was equalized by subsampling the trials in a histogram-matching procedure with 200 ms bins. This resulted in using only ~50% of the trials. The same analysis was then applied to this reaction time matched dataset.

## Group analysis

To account for both within-subject and across-subject variance, statistical testing of the electrophysiological data was conducted with multilevel models (*Aarts et al., 2014*; *Goldstein, 2011*) (also known as random effect models). Random factors included electrodes nested within subjects. Significance of interactions and/or main effects was assessed with a likelihood ratio test against a null model excluding that particular term.

For comparison of latency across regions, we restricted our analyses to simultaneous measurements made within each subject. We computed the latency difference for each pair of

simultaneously recorded electrodes from different regions. The F-statistic of this latency difference across the groups was compared against a null distribution generated by shuffling, within each subject, the region labels (n = 10,000 shuffles). Post hoc testing used the Benjamin-Hochberg procedure to control for multiple comparisons.

## Cross-frequency coupling

To measure cross-frequency coupling between the theta and gamma frequency bands, we used the Modulation Index (MI) defined previously (*Tort et al., 2008*). Activity in the theta (4–8 Hz) and high gamma (70–120 Hz) bands was obtained with a zero-phase least-squares finite impulse response (FIR) filter. Instantaneous phase and amplitude was extracted with the Hilbert Transform. For the Stroop and Reading Task separately, the MI was computed as the Kullback-Leiber distance between the phase-amplitude histogram and a uniform distribution. For comparison between tasks, the number of trials was equalized. This MI was compared against a surrogate distribution generated by randomly lagging the time series across 1000 repetitions. Similar results were obtained with the measure defined in Canolty et al. (*Canolty et al., 2006*). Results were also similar when a surrogate distribution was created by randomly pairing low-frequency phase with high-frequency power from different trials.

To compare the strength of cross-frequency coupling between congruent and incongruent conditions, we computed the difference in MI between the two conditions while equalizing the trial count. This difference was compared against a null distribution generated by randomly shuffling the congruent and incongruent labels.

## Acknowledgements

We thank all the patients for participating in this study, Sheryl Manganaro, Jack Connolly, Paul Dionne, and Karen Walters for technical assistance. We thank Ishita Basu for assistance in performing experiments. We thank Sam Gershman for comments on the manuscript. This work was supported by NIH (DP2OD006461) and NSF (0954570 and CCF-1231216).

## Additional information

### Funding

| Funder | Grant reference number | Author |
| --- | --- | --- |
| NIH Blueprint for Neuroscience Research | | Gabriel Kreiman |
| National Science Foundation | 1358839 | Gabriel Kreiman |
| National Science Foundation | CCF-1231216 | Gabriel Kreiman |

The funders had no role in study design, data collection and interpretation, or the decision to submit the work for publication.

### Author contributions

HT, Conception and design, Acquisition of data, Analysis and interpretation of data, Drafting or revising the article; H-YY, C-CC, NEC, Performed the experiments, Commented and approved the manuscript, Acquisition of data; JRM, Performed the neurosurgical procedures, Commented and approved the manuscript, Acquisition of data; WSA, Performed the neurosurgical procedures, Commented and approved the manuscript, Conception and design, Acquisition of data; GK, Conception and design, Analysis and interpretation of data, Drafting or revising the article

### Author ORCIDs

Gabriel Kreiman, http://orcid.org/0000-0003-3505-8475

### Ethics

Human subjects: Informed consent, and consent to publish, was obtained for each participant. All procedures were approved by the Institutional Review Boards at each institution (Methods).

# Additional files

## Supplementary files

• Supplementary file 1. Information about each of the 15 subjects that participated in this study

• Supplementary file 2. Information about each of the 51 electrodes that were modulated by conflict.

• Supplementary file 3. Distribution of the 51 electrodes that were modulated by conflict across the four frontal cortex areas and the 15 subjects in our study.

• Supplementary file 4. Statistical reporting table describing the statistical tests used throughout the text, including the corresponding n, the number of degrees of freedom, p values and effect sizes.

• Source code 1. Code to plot *Figure 2A* (with minor modifications, this code can also be used to plot the other example figures in the text).

• Source code 2. Code to plot *Figure 4*.

• Source code 3. Code used to plot error bars in the plot Figure2A.

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
