## [Decision Letter]

Thank you for submitting your work entitled "Cascade of neural processing orchestrates cognitive control in human frontal cortex" for consideration by *eLife*. Your article has been favourably evaluated by David Van Essen (Senior Editor) and three peer reviewers, one of whom, Hiram Brownell, is a member of our Board of Reviewing Editors.

The reviewers have discussed the reviews with one another and the Reviewing Editor has drafted this decision to help you prepare a revised submission.

Summary:

This paper reports a human ECoG study, which provides evidence for conflict-related signals in dorsal anterior cingulate cortex, and shows that such signals propagate subsequently to dorsolateral and orbitofrontal cortex. The results are important for two reasons: (1) They add to another recent human intracranial study (by Sheth and colleagues) in strongly contradicting non-human primate studies, which have – at least until recently – found no evidence for conflict signaling in monkey cingulate, despite the apparent presence of such signals in a large number of human fMRI studies; (2) they provide some of the strongest evidence yet that conflict signals arise earliest in dorsal cingulate, with dorsolateral PFC following later. This finding is consistent with a recent crop of theoretical perspectives on frontal lobe function, which have generally placed cingulate in the role of recruiting dorsolateral PFC when there is a demand for cognitive control.

Essential revisions:

1) Add clarity about how electrodes were chosen to go into the analysis of the temporal hierarchy of conflict related signals. For example, Figure 6 says there are 36 electrode pairs between DLPFC and OFC, yet Figure 3 says there are only 24 electrodes in DLPFC that showed a significant difference in γ power between incongruent and congruent trials. Is the latter a prerequisite to be included in the former? The concern is that the peak timing in one or two electrodes might then become particularly influential in determining the calculated difference in timings. The authors should clarify whether this is a problem, and how they dealt with it when computing the statistics on latency difference.

2) Provide more detail about the distribution of significant electrodes in each individual subject for each region. (This will make clear whether electrodes were only clustered in a few subjects or evenly distributed across the population, and also make clear where the significant pairs of electrodes came from).

3) One reviewer suggested that it would make more sense to test/plot the logarithm of the γ power ratio when comparing the incongruent and congruent responses (Figure 4/D/E). The ratio, if not log-transformed, is biased towards having values that are pushed further away from 1 if the numerator is larger than the denominator rather than vice versa (i.e. if incongruent power is double that of congruent, the ratio is 2; if congruent power is double incongruent, the ratio is 0.5). The log (ratio), by contrast, would show the same deviation from zero for an equivalent change in either numerator or denominator.

4) The OFC is said to show variable latencies across subjects, causing a flatter-looking response profile than other areas (Figure 4). Might this be because the OFC is more sensory-evoked than other areas recorded? It looks a bit like this from the example electrode, but of course there will be substantial heterogeneity across recordings. Consider adding including as a supplement helpful the sensory-locked equivalent of Figure 4 for all regions, as a supplement.

5) Earlier work by Nakamura and Olson in 2005 (J Neurophys 93, 884-908) looked at apparent conflict responses in monkey supplementary eye fields, and concluded that such signals were essentially epiphenomenal. The neurons they examined all showed response-selective activity, and the 'conflict' activity seen at the population level turned out to reflect non-linear modulation of response-related activity when responses competed. This always struck me as a very serious problem for the 'conflict monitoring' theory. The question is how the present authors can rule out this kind of explanation for their own results. The second paragraph of the subsection “Conflict responses in frontal cortex” reports that γ-band power was invariant to the particular word-color combination presented, which makes it sound like the neural responses being recorded were not response specific. If this were indeed the case, then it would clearly rule out a Nakamura-Olson scenario. And this conclusion would be possible if the data being reported involved single-neuron recordings. However, it is not clear that the reported findings, in an ECoG setting where you're seeing population-level effects, can rule out this alternative interpretation.

---

## [Author Response]

*1) Add clarity about how electrodes were chosen to go into the analysis of the temporal hierarchy of conflict related signals. For example, Figure 6 says there are 36 electrode pairs between DLPFC and OFC, yet Figure 3 says there are only 24 electrodes in DLPFC that showed a significant difference in γ power between incongruent and congruent trials.Is the latter a prerequisite to be included in the former? The concern is that the peak timing in one or two electrodes might then become particularly influential in determining the calculated difference in timings. The authors should clarify whether this is a problem, and how they dealt with it when computing the statistics on latency difference.*

We thank for the reviewers for bringing up this important point. We first determined whether the neural responses were modulated by conflict and obtained n=51 electrodes across our electrode population (paragraph four, subsection “Conflict responses in frontal cortex”). From this subset of electrodes, we considered all possible pairs of *simultaneously recorded electrodes* across regions (i.e. within the same subject, subheading “Regional differences in conflict response latencies”). This requirement significantly lowers the number of possible pairs: electrode locations are purely dictated by clinical reasons and we did not always have simultaneous pairs for all possible combinations of regions. It is difficult to derive the number of electrode pairs in Figure 6 (e.g. 36 in the question above) from the number of electrodes in Figure 3 (e.g. 24 in the question above). We have now included a new table ([Supplementary-material SD9-data]) to clarify this, describing the distribution of the 51 electrodes modulated by conflict in each subject. To follow up on the specific question above, there are 24 electrodes in DLPFC. The number 36 is derived from 7 subjects: 3x3+5x3+1x1+1x1+1x3+1x1+6x1.

The ACC-PFC latency difference is based on one ACC electrode and six PFC electrodes (a total of 6 pairs). We emphasize this point in the text (paragraph one, subheading “Regional differences in conflict response-latencies”). There is only one electrode pair between ACC and OFC and we opted not to report this in Figure 6 because it is only a single case. All the other region pairs we examined have contributions from multiple electrodes across multiple patients.

For the statistical analysis, we first assessed whether the average latency difference differed significantly across the four regions (non-parametric ANOVA). We generated a null distribution by randomly shuffling the region labels within each subject and recomputing the average latency difference between all possible pairs. This procedure preserves both the statistical properties of each electrode and the number of pairs. Thus, if there were few electrode pairs for a given comparison, the same few pairs would also be driving the latency differences in the null distribution. We assessed the statistical significance of latency differences in each region pair by comparing against the null distribution. This is discussed in the Methods, paragraph two, subheading “Group Analysis.”.

*2) Provide more detail about the distribution of significant electrodes in each individual subject for each region. (This will make clear whether electrodes were only clustered in a few subjects or evenly distributed across the population, and also make clear where the significant pairs of electrodes came from).*

[Supplementary-material SD7-data] reports the total number of electrodes per subject. [Supplementary-material SD8-data] reports detailed response metrics for each of the electrodes modulated by conflict in each subject. We have now added [Supplementary-material SD9-data] with a clearer description of the distribution of significant electrodes in each individual subject. The distribution of electrodes modulated by subject as well as the corresponding pairs can be more directly read from this Table. Because electrode locations are determined by the clinical needs, the coverage varies from subject to subject. To account for this subject-to-subject variation, we used a multilevel model in our group analysis, as noted in paragraph five, subheading “Conflict responses in frontal cortex” and subsection “Group Analysis”.

*3) One reviewer suggested that it would make more sense to test/plot the logarithm of the γ power ratio when comparing the incongruent and congruent responses (Figure 4/D/E). The ratio, if not log-transformed, is biased towards having values that are pushed further away from 1 if the numerator is larger than the denominator rather than vice versa (i.e. if incongruent power is double that of congruent, the ratio is 2; if congruent power is double incongruent, the ratio is 0.5). The log (ratio), by contrast, would show the same deviation from zero for an equivalent change in either numerator or denominator.*

We have replaced all the relevant panels of Figure 4 (as well as Figure 4—figure supplement 1 and the new Figure 4—figure supplement 3) with log-transformed ratios. We note that the ratios are used for display purposes only, and the statistical analyses with the multi-level model use the raw γ power of each condition individually (Figure 4 legend). Therefore, this change does not modify any of our statistical tests.

Based on this question, but unrelated to Figure 4, we also realized that the same issue pointed out by the reviewers here could be subtly observed in our single electrode examples, particularly before t=0 in the stimulus aligned-responses, (e.g. Figure 2), where we had normalized the responses by dividing by the baseline and then averaged them. While the statistical analyses were computed on the raw responses without normalization and therefore this point did not affect any of the statistics or conclusions, some of the previous figures showed a slightly above 1 response before t=0. Therefore, we have now replaced these displays with the z-scored γ power in Figure 2, Figure 2—figure supplement 1, Figure 2—figure supplement 2, and Figure 5.

*4) The OFC is said to show variable latencies across subjects, causing a flatter-looking response profile than other areas (Figure 4). Might this be because the OFC is more sensory-evoked than other areas recorded? It looks a bit like this from the example electrode, but of course there will be substantial heterogeneity across recordings. Consider adding including as a supplement helpful the sensory-locked equivalent of Figure 4 for all regions, as a supplement.*

We have included a supplement to Figure 4 (see Figure 4—figure supplement 3) with sensory-locked averages across regions. From this figure averaging across electrodes, it does not seem that there is a stronger sensory evoked response in the OFC that is selective for conflict.

*5) Earlier work by Nakamura and Olson in 2005 (J Neurophys 93, 884-908) looked at apparent conflict responses in monkey supplementary eye fields, and concluded that such signals were essentially epiphenomenal. The neurons they examined all showed response-selective activity, and the 'conflict' activity seen at the population level turned out to reflect non-linear modulation of response-related activity when responses competed. This always struck me as a very serious problem for the 'conflict monitoring' theory. The question is how the present authors can rule out this kind of explanation for their own results. The second paragraph of the subsection “Conflict responses in frontal cortex” reports that γ-band power was invariant to the particular word-color combination presented, which makes it sound like the neural responses being recorded were not response specific. If this were indeed the case, then it would clearly rule out a Nakamura-Olson scenario. And this conclusion would be possible if the data being reported involved single-neuron recordings. However, it is not clear that the reported findings, in an ECoG setting where you're seeing population-level effects, can rule out this alternative interpretation.*

We thank the reviewers for raising this important question. It is indeed the case that the neural responses where we observe conflict-selective signals were not response-selective. We do observe sensory responses selective to color or word along the ventral visual stream as well as motor selective responses in motor cortex (now mentioned briefly in paragraph three, Results). However, out of the n=51 conflict- selective electrodes in frontal cortex, we observe response-specific signals in only 1 electrode. Thus, *at the level of EcoG field potentials*, the modulation by conflict cannot be simply explained on the basis of selectivity to specific color/word combinations.

Unfortunately, in this set of patients, we did not have access to single neuron recordings. Although some investigators have claimed that γ power in field potential signals reflects population-firing rates from the underlying neuronal populations (Manning et al., 2009; Ray and Maunsell, 2011), the relationship between spikes and field potentials is more complex (Buzsaki et al., 2012). We cannot rule out the possibility that individual neurons in these areas show stronger selectivity to specific color/word combinations and that this is averaged out in our signals. We now discuss this point in paragraph five, Discussion.

Additionally, we emphasize that the γ power signals that we observe correlates with several elements of behavior including correlation with reaction time (Figure 2 and Figure 4), reflection of the Gratton effect (Figure 4) and detection and correction of error trials (Figure 5). These correlates suggest that the signals we measure are not just an epiphenomenon.